# Catalytic Oxidation of Benzoins by Hydrogen Peroxide on Nanosized HKUST-1: Influence of Substituents on the Reaction Rates and DFT Modeling of the Reaction Path

**DOI:** 10.3390/molecules28020747

**Published:** 2023-01-11

**Authors:** Darya V. Yurchenko, Anton S. Lytvynenko, Emir N. Abdullayev, Nina V. Peregon, Konstantin S. Gavrilenko, Alina O. Gorlova, Sergey V. Ryabukhin, Dmitriy M. Volochnyuk, Sergey V. Kolotilov

**Affiliations:** 1L.V. Pisarzhevskii Institute of Physical Chemistry of the National Academy of Sciences of Ukraine, Prosp. Nauky 31, 03028 Kyiv, Ukraine; 2Department of Analytical Chemistry, Faculty of Science, Charles University, Albertov 6, 12800 Prague, Czech Republic; 3Enamine Ltd., 78 Chervonotkatska Str., 02094 Kyiv, Ukraine; 4Institute of High Technologies, National Taras Shevchenko University of Kyiv, 60 Volodymyrska Str., 01033 Kyiv, Ukraine; 5Institute of Organic Chemistry of the National Academy of Sciences of Ukraine, 5 Murmanska Str., 02094 Kyiv, Ukraine

**Keywords:** HKUST-1, benzoin, hydrogen peroxide, oxidation, catalysis, adsorption, DFT calculations

## Abstract

In this research, the oxidation of a series of benzoins, R-C(=O)-CH(OH)-R, where R = phenyl, 4-methoxyphenyl, 4-bromophenyl, and 2-naphthyl, by hydrogen peroxide in the presence of nanostructured HKUST-1 (suspension in acetonitrile/water mixture) was studied. The respective benzoic acids were the only products of the reactions. The initial average reaction rates were experimentally determined at different concentrations of benzoin, H_2_O_2_ and an effective concentration of HKUST-1. The sorption of the isotherms of benzoin, dimethoxybenzoin and benzoic acid on HKUST-1, as well as their sorption kinetic curves, were measured. The increase in H_2_O_2_ concentration expectedly led to an acceleration of the reaction. The dependencies of the benzoin oxidation rates on the concentrations of both benzoin and HKUST-1 passed through the maxima. This finding could be explained by a counterplay between the increasing reaction rate and increasing benzoin sorption on the catalyst with the increase in the concentration. The electronic effect of the substituent in benzoin had a significant influence on the reaction rate, while no relation between the size of the substrate molecule and the rate of its oxidation was found. It was confirmed by DFT modeling that the reaction could pass through the Baeyer–Villiger mechanism, involving an attack by the HOO^−^ anion on the C atom of the activated C=O group.

## 1. Introduction

In the last decades, a lot of attention has been paid to heterogeneous catalysts for oxidation reactions of organic compounds [1,2,3]. The advantages of such systems include the simple separation of the products from the catalyst and the possible use of the catalysts in flow reactors, as well as the possibility to perform the oxidation selectively due to the size discrimination of the reactants [4,5,6]. Porous coordination polymers (PCPs), also known as metal–organic frameworks (MOFs), are promising compounds for such applications because they have the highest, among the metal-containing heterogeneous catalysts of other classes, volume concentration of accessible metal ions in combination with their regular porosity (required for size selectivity) [7,8], their ability to tune the electronic structure of the metal sites [9,10], and their ability to incorporate special functional groups (such as chiral [11,12] or basic sites [13,14] or Brønsted acidic sites [15] in addition to metal ions, which act as Lewis acids [16]), as well as their centers with special properties (such as redox-active ones [17,18,19]).

As described above, catalytically active sites can form in the pores of porous coordination polymers, but their accessibility can be limited by diffusion obstacles [20,21]. The use of nanoparticles of these coordination polymers as catalysts can be a solution of this problem, because higher fraction of the catalytically active sites becomes accessible [22,23]. On the other hand, this approach would lead to a loss of selectivity of the PCPs, associated with size discrimination, due to the increased contribution of the active sites located on the surface of PCP particles in the catalytic process (where the metal ions are accessible to the molecules of any size). This notwithstanding, nanostructured PCPs preserve many of the advantages of the catalysts of this class, such as high activity and the capacity to use these compounds as heterogeneous catalysts for flow reactors [24,25,26].

Copper(II) 1,3,5-benzenetricarboxylate, with the formula Cu_3_(btc)_2_(H_2_O)_3_ (known as HKUST-1 [27], where btc^3−^ = 1,3,5-benzenetricarboxylate), contains a 3D system of pores with ca. 1 nm size, which are potentially accessible to many organic molecules [27]. Water molecules coordinated to Cu^II^ ions can be easily substituted by other compounds in solutions [28]. Notably, the thermal activation of HKUST-1 prior to catalytic experiments, often used for the elimination of the coordinated water molecules, is of disputable importance in many cases, because water either forms in the reaction (normally, water forms in the catalytic reaction in close proximity of the Cu^II^ ion and should be readily coordinated by such an ion) or the reaction is carried out in a water-containing solvent or in the solvent, which can coordinate to the Cu^II^ ions itself. Due to these features, HKUST-1 combines the presence of easily accessible Cu^II^ ions along with relatively high hydrolytic stability and low price, which is important for the use of this compound as a catalyst in organic chemistry.

It was shown that HKUST-1 has a high catalytic activity in many oxidation processes, such as the oxidation of *trans*-ferrulic acid into vanillin by H_2_O_2_ [29], the hydroxylation of benzene by H_2_O_2_ [30], the oxidation of benzylic compounds with *t*-butylhydroperoxide [31], or the oxidation of benzylic alcohols by oxygen in air (in combination with a TEMPO co-catalyst) [32]. The mechanisms of these important processes were not studied in detail, and both ionic and free radical oxidation could be proposed as possible options for the oxidation of organic compounds by peroxides catalyzed by HKUST-1 (different mechanisms for different reactions).

The aim of this work was to elucidate the specific features of the catalytic oxidation of benzoin and substituted benzoins by hydrogen peroxide in the presence of HKUST-1. The reaction of oxidation of benzoins as selected because of the high importance of alcohol oxidation in mild conditions for fine organic synthesis [33,34]. It was supposed that the oxidation of α-hydroxyketones could pass through the Baeyer–Villiger mechanism [35], which involves a nucleophilic attack of a peroxide anion both on benzoin (the compound containing the –CH(OH)-C(=O)- fragment) and benzil (the product of benzoin oxidation, i.e., the compound with the –C(=O)-C(=O)- moiety). Both these pathways should finally result in the formation of the same products, i.e., benzoic acids, but involve different intermediates and reactions after the Baeyer–Villiger rearrangement step. It was shown that the acidic sites (both Brønsted [36] and Lewis [37,38,39,40,41]) of the catalyst were required for the Baeyer–Villiger oxidation, and the Cu^II^ ions in HKUST-1 could be considered as suitable catalytic centers, because their acidic nature was proven experimentally [42,43]. Despite the large array of data on the oxidation of benzoins with nitric acid or nitrates, information on the direct oxidation of benzoin with H_2_O_2_ is quite scarce [44,45,46], and the quantity of reports on the catalytic activity of Cu^II^ species in the Baeyer–Villiger oxidation processes is also very limited [47,48].

Hydrogen peroxide was chosen as the oxidant because it is considered as the most “green” oxidant, which forms water as the only by-product [49,50,51]. However, many reactions involving H_2_O_2_ lead to the formation of complex mixtures of products, and this observation gave the grounds to suppose that a mechanism of uncontrolled oxidation by radical occurs [52,53]. Studies of the specific features of the oxidation of organic compounds by hydrogen peroxide in the presence of new catalysts is an important task.

In this paper, we report the experimental results obtained on the oxidation of a series of benzoins bearing different substituents by hydrogen peroxide in the presence of HKUST-1, as well as the results of DFT modeling of the reaction.

## 2. Results and Discussion

### 2.1. Synthesis and Characterization of Nanosized HKUST-1

The synthesis of HKUST-1 was performed by electrochemical dissolution of a copper anode in a solution of H_3_btc (see Experimental section for details) following a modification of the procedure reported in [54]. The identity and phase purity of the sample were confirmed by comparison of the results of XRD measurements with a reference XRD pattern calculated for the reported HKUST-1 crystal structure [27] using the CCDC Mercury software [55] (Figure 1). It was also shown that the sample did not contain other crystalline phases.

TEM images revealed that the synthesized samples of HKUST-1 consisted of aggregated nanoparticles of an irregular shape (Figure 2a). The separate particles were dozens of nanometers in size, and the aggregates were hundreds of nanometers in size. The particles of HKUST-1 looked like semi-transparent blocks on the image, which was consistent with the fact that this compound is built primarily by light elements such as carbon and hydrogen. This feature allowed the determination of the boundaries of the separate particles within the aggregates by comparing the transparency level (shades of gray) of various parts of the aggregates. Thus, the size of the particles was measured, and their distribution was presented as histogram with 10 nm intervals (Figure 2b). The maximum of this distribution was centered at 40–60 nm, and this value was consistent with the size of the crystalline particles (ca. 40–45 nm) determined from XRD by Selyakov–Sherrer’s equation [56], as described previously [41].

The porosity of HKUST-1 was characterized via measurements of the nitrogen ad(de)sorption isotherm (Figure 3). The BET specific surface area was S_BET_ = 1435 cm^2^/g, and the micropore volume calculated using Dubinin–Radushkevich model was V_DR_ = 0.61 cm^3^/g. This value, as well as the entire shape of the isotherm, were consistent with the ones reported for HKUST-1 samples prepared by other methods [54,57] (moreover, the value of S_BET_ was closer to the higher boundary of the range of the specific surface areas reported for HKUST-1 prepared by different methods). The obtained HKUST-1, with an average particle of size 40–60 nm, was used for the experiments on catalytic benzoin oxidation.

### 2.2. Catalytic Oxidation of Benzoin and Analogs by H_2_O_2_

It was found that benzoic acid was the only product of benzoin oxidation with a ca. 100-fold excess of hydrogen peroxide (Figure 4). This finding was consistent with the previously reported results on the catalytic oxidation of benzoin with a three-fold excess of H_2_O_2_, which led to a mixture of benzil and benzoic acid (or its esters) [58,59,60,61,62] and the cleavage of benzoin by oxone with the formation of an ester of benzoic acid with a more than 70% yield [35].

It was found that the dependency of the benzoin concentration (*c*) on time (*t*) during the first 20–30 min of the reaction could be approximated by a straight line (Appendix A). Thus, the method of initial rates could be used for a comparison of the process rates observed for the different benzoin structures and different experimental conditions. The average initial rates were determined as the slope of the dependency of *c* vs. *t* at different concentrations of the reagents and effective concentrations of the catalyst.

In order to confirm that the catalytic action was caused by HKUST-1 but not by the Cu^2+^ ions in solution, several tests were carried out. First, it was shown that the X-ray diffraction patterns of HKUST-1 before and after the catalytic experiments were identical (Figure 1). Second, a catalyst filtration test was carried out; after the separation of the solid catalyst from a part of the reaction mixture, the rate of benzoin oxidation became lower, which was similar to the rate of the non-catalytic process. Third, no Cu^2+^ ions could be detected in all the reaction mixtures by qualitative testing (water solution of KI/starch probe) after filtration of HKUST-1. In addition, for the experiments where the highest reaction rates were observed, the concentrations of Cu^2+^ in the filtered reaction mixtures, measured by atomic adsorption spectroscopy, were less than 10^−7^ M (at least 1000 times lower than the lowest effective concentration of HKUST-1 used in the catalytic experiments).

Both the average initial rate of benzoin oxidation and the conversion of benzoin measured after 3 h of reaction (which reflected average reaction rate in this time period) linearly grew with the increase in H_2_O_2_ concentration (Figure 5). Thus, it can be concluded that the reaction was first order with respect to H_2_O_2_.

The oxidation of benzoin by H_2_O_2_ takes place without addition of a catalyst (the point at zero c_eff._(HKUST-1) in Figure 6a corresponds to the rate of non-catalytic oxidation). The dependency of the average initial reaction rate on the initial effective concentration of HKUST-1 (suspension in the reaction mixture) passed through a maximum (Figure 6a) at c_eff._(HKUST) ca. 1.3 mmol/l (hereinafter the effective concentration of HKUST-1 is expressed per 1 Cu^2+^ ion). A further increase in HKUST-1 loading led to a significant decrease in the initial reaction rate, and at c_eff._(HKUST-1) = 2.0 mmol/l, it was close to the rate of non-catalytic oxidation. Similarly, the values of benzoin conversion after 3h vs. the effective concentration of HKUST-1 in the reaction mixture also passed through a maximum at c_eff._(HKUST-1) ca. 1.0 mmol/l and drastically decreased upon an increase in c_eff._(HKUST-1). Such a tendency can be explained by the interplay of the two factors: (1) acceleration of the reaction by the catalyst and (2) the decrease in the benzoin concentration due to its adsorption on the catalyst, provided that the “captured” benzoin could not participate in the reaction. The adsorption of benzoin on HKUST-1 was quantitatively characterized (vide infra); briefly, 1 mmol of HKUST could absorb ca. 6 mmol of benzoin, causing a ca. 20% decrease in its concentration in the solution in the conditions of the experiment.

In order to verify that the decrease in the benzoin oxidation rate upon the increase in the effective HKUST-1 concentration was not caused by the catalytic decomposition of H_2_O_2_, in a separate experiment, the concentration of H_2_O_2_ in the presence of HKUST-1 was monitored by titration of the aliquots with KMnO_4_ solution. It was found that the concentration of H_2_O_2_ did not change significantly during 3 h, thus, the catalytic decomposition of H_2_O_2_ could not be the reason for the above-mentioned change in the benzoin oxidation rate.

The dependency of the average initial oxidation rate and the conversion after 3 h on the initial benzoin concentration also passed through a maximum (Figure 7), similarly to the dependencies of these values on the effective HKUST-1 concentrations described above. Such a change in the reaction rate can be explained by the blocking of the catalytically active sites (which prevented access by H_2_O_2_) because of adsorption of the substrate. Notably, the adsorption of the reaction product—benzoic acid—on the HKUST-1 was also significant, vide infra, but it could not affect the initial reaction rate measured at the time period when conversion of benzoin was small.

Since it was supposed that adsorption of benzoin on the catalytic sites of HKUST-1 was the main reason for the suppression of the catalytic action of HKUST-1, the adsorption of this compound on the HKUST-1 (a sample from the same batch as used for the catalytic experiments) was measured. The shape of the benzoin adsorption isotherm (Figure 8) was not typical for adsorption on microporous sorbents. The adsorption of benzoin was negligibly small at concentrations less than ca. 0.01 M, while an increase in the concentration led to an abrupt growth of the isotherm. The quantity of absorbed benzoin quickly exceeded the value expected for filling the pores of HKUST-1; the pore volume estimated from N_2_ sorption was V_DR_ = 0.61 cm^3^/g, and the quantity of benzoin that can be located in such pores was ca. 0.8 g per 1 g of HKUST-1. Such a shape of the adsorption isotherm can be explained by the multi-layer sorption of benzoin on the HKUST-1 nanoparticles, both in the pores and on the external surface of the nanoparticles.

The conclusion about the multi-layer adsorption was consistent with the results of the kinetic experiments, i.e., a decrease in the average initial oxidation rate upon an increase in the effective HKUST-1 concentration—the addition of a larger quantity of HKUST-1—led to a significant decrease in the benzoin concentration in solution. At the same time, some portion of the HKUST-1 probably did not participate in the catalytic oxidation due to blocking of the active sites. This conclusion was consistent with the observed decrease in the oxidation rate upon an increase in the benzoin concentration, which could be associated with the blocking of the catalyst. For illustration purposes, the quantity of absorbed benzoin is shown for different concentrations of this compound vs. the average initial oxidation rate in Figure 9.

The reaction product—benzoic acid—could also block the pores of the HKUST-1 catalyst upon its accumulation in the reaction mixture, slowing down further benzoin oxidation. In order to estimate the extent of such pore blocking and to make a conclusion regarding its influence on the benzoin oxidation, the isotherm of benzoic acid sorption on HKUST-1 was measured (Figure 10). It was found that the sorption capacity of the HKUST-1 sample with respect to benzoic acid significantly exceeded the accessible pore volume (the pores could accommodate ca. 0.8 g of the acid per 1 g of the catalyst). Similarly, as in the case of benzoin adsorption, the higher sorption capacity of HKUST-1 with respect to the benzoic acid compared to the value expected for complete pore filling can be explained by multi-layer benzoic acid sorption on the external surface of the nanoparticles. Thus, the sorption of the reaction product could be a factor suppressing the catalytic activity of the HKUST-1 in the process of benzoin oxidation. In view of this experimentally proven significant sorption of the benzoic acid on HKUST-1, it was difficult to estimate if there was any contribution from its acidity (the continuous increase in the concentration of the benzoic acid in the reaction mixture) on the average initial rate of benzoin oxidation. If the first step of the reaction involves participation of a HOO^−^ anion (as it should be in the case of the Baeyer–Villiger reaction pathway), it can be supposed that an increase in the benzoic acid concentration should slow down the oxidation reaction because of the suppression of H_2_O_2_ dissociation.

The sorption of benzoin and benzoic acid reached saturation after ca. 200 s (Figure 11), which was quite quick in the time scale of the oxidation process (in the majority of the experiments where c(H_2_O_2_) was 3.3 M, the benzoin conversion after 3 h of reaction did not exceed 60%). It can be concluded that the rate of benzoin or benzoic acid sorption did not have a significant contribution to the rate of benzoin oxidation.

The kinetic curve of benzoin sorption can be fitted with the pseudo-second-order equation (Equation (1)), with the parameters *a*_max_ = 0.012 mole/g, *k*_2_ = 8.0 (g of HKUST)(mole of benzoin)^−1^s^−1^ or 1.2 (mole of HKUST)(mole of benzoin)^−1^(min)^−1^ for c_0_(benzoin) = 0.0165 M.
(1)$$\frac{t}{a_{t}}=\frac{1}{k_{2}a_{\max}^{2}}+\frac{t}{a_{\max}}$$

The observation that the benzoin sorption kinetics were adequately described by the pseudo-second-order model could be an indication that the binding of benzoin was the sorption-rate-limiting step (in contrast to the diffusion of this reactant to the surface of the HKUST-1 particles) [63]. This finding was consistent with the conclusion regarding the multi-layer sorption on the nanoparticles of the HKUST-1.

In order to study the influence of the benzoin structure on the efficiency of the oxidation process, the yields of the corresponding acids and the average initial reaction rates were studied for a series of substituted benzoins (Figure 12). The highest values of the initial average oxidation rate and the highest conversion (after 3 h of reaction), both for the catalytic and non-catalytic reactions, were found for 4,4′-dibromobenzoin. The lowest values were observed for 4,4′-dimethoxybenzoin.

In order to check if the lowest reaction rate for the oxidation of 4,4′-dimethoxybenzoin was not caused by its exceptionally high adsorption and the efficient blocking of the catalytically active sites, the isotherm of the adsorption of this compound on the HKUST-1 and the kinetic curve of adsorption were measured (Figure 13).

It was found that the sorption capacity of HKUST-1 with respect to 4,4-dimethoxybenzoin was lower compared to the unsubstituted benzoin, though the former value exceeded the one expected for filling the accessible pores of HKUST-1, which was similar to the case for benzoin sorption. The shape of the isotherm of 4,4′-dimethoxybenzoin sorption was significantly different to the shape of the isotherms of benzoin or benzoic acid sorption—the former reached saturation at ca. 0.015 mole/g. The sorption equilibrium in the process of 4,4′-dimethoxybenzoin sorption was reached after ca. 300 s, which was comparable with the rate of benzoin sorption.

Thus, the low rate of 4,4-dimethoxybenzoin oxidation was caused by its electronic structure rather than by the blocking of the catalyst. Notably, steric hindrances also seemed to not be the critical factor that controlled the reaction rate, because the bulkiest benzoin—the one which contained β-naphthyl substituents instead of phenyl—showed one of the highest oxidation rates in the studied series.

### 2.3. Analysis and DFT Calculations of the Reaction Pathway

Two principally different mechanisms can be proposed for the reaction of benzoin oxidation: the attack of the HOO^−^ anion on the C atom of the C=O group, and the attack of the •OH radical on the C(=O)-CH(OH) fragment. The efficiency of the HOO^−^ attack on the C atom of the C=O group, bearing a partial positive charge, should increase upon the introduction of an electron-withdrawing group (because it increases the partial positive charge on the C atom), and it should decrease upon the introduction of an electron-donating group. On the other hand, the efficiency of the •OH radical attack on the molecule should not significantly depend on the charge distribution in the molecule. The dependency of the benzoin oxidation reaction rate on the nature of the substituents in the phenyl rings, consistent with the theoretically expected one, could be an argument for the reaction mechanism, involving an attack of the peroxide anion (in contrast to the •OH radical) on the C atom of the carbonyl group.

In order to verify this hypothesis, the atomic charges of the C atoms of the carbonyl groups in the substituted benzoins considered in this study were estimated in an attempt to find any correlation between the initial oxidation rate and such a charge. Such a correlation would exist if the attack of the peroxide anion was the rate-limiting step of the whole oxidation. Such an estimation of the atomic charge on the carbon atom of the carbonyl group did not reveal any notable differences between the benzoins considered herein (Appendix A). While the charges determined for one compound by different methods (relying on different principles) varied up to 0.45 (in terms of elementary charges), the difference in the charges determined for different compounds within one method typically did not exceed 0.05. The only deviation from this behavior was with the modified Becke approach (details of calculation are presented in the Experimental section); the charges determined within this method ranged from 0.23 in the unsubstituted benzoin to 0.46 in the naphthyl-substituted one. However, this difference could be explained as an artifact caused by the arbitrary nature of the Becke partitioning function (the function which “distributes” the spatial electronic density in the molecule between the atoms).

For the case of radical oxidation, the geometric optimization of a species containing a benzoin molecule and an •OH radical resulted in the barrierless abstraction of the hydrogen atom attached to the carbon atom of the >CH-OH group of the molecule (Figure 14).

It can be concluded that the radical oxidation of benzoin could easily occur; however, the source of the radical is required. It should be noted that no decomposition of H_2_O_2_ in the presence of HKUST-1 was found in this study, vide supra. Thus, we excluded radical oxidation because the formation of the •OH radical did not find any experimental proof in this study. The possible radical mechanism of the oxidation of α-diones by H_2_O_2_ was previously excluded [64]. The role of radicals in the course of the oxidation of different compounds by H_2_O_2_ was also considered to be doubtful in another study [65].

The modification of the Baeyer–Villiger oxidation mechanism was another possible reaction pathway for the first stage of benzoin oxidation, leading to C-C bond cleavage. Such a mechanism involves the addition of a HOO^−^ anion to the carbonyl compound followed by a rearrangement of the intermediate. In order to verify if Baeyer–Villiger oxidation could occur, a DFT modeling of the reaction path and key intermediates was carried out. It was shown previously that DFT is an efficient tool for modeling the structures of molecules [66,67,68] and intermediates [69] in the reactions. For the DFT modeling in this case, the carbonyl group was activated by coordination to the 6^th^ position in the coordination sphere of the Cu^2+^ ion within the Cu_2_ dimers in the HKUST-1 (such a coordination occurred in a barrierless way according to our DFT calculations, assuming that the position was free from an acetonitrile molecule). An addition of HOO^−^ to the system did not result in a barrierless bond formation (Figure 15a). However, its nudged movement toward the carbon atom of the carbonyl group (changing the distance between the carbon atom C8 and the deprotonated oxygen atom O60 of HOO^−^, Figure 15a) from the equilibrium value of 2.67 to the value of 1.27 Å, which was slightly below the sum of their covalent radii, 1.39 Å [70], via the relaxed surface scan procedure (RSS, see Experimental part for details of the approach) showed only minimal barrier of 2–3 kJ/mol (Figure 16a). Such a barrier could be easily overcome by thermal motion (the final geometry of the RSS procedure result subjected to a consequent unconstrained optimization is presented in Figure 15b). A further step, considered as a limiting stage of the Baeyer–Villiger oxidation, implied the movement of the PhCHOH substituent (namely the C10 atom) from the C8 atom to the O60 atom. Within our DFT modeling, this movement was enforced by the gradual shortening of the distance between C10 and O60 from the equilibrium value of 2.364 to 1.364 Å. This process, without any other constraints, resulted in the rearrangement product depicted in Figure 15c. The maximum of the RSS energy curve (Figure 16b) corresponding to a transition state with an epoxy-type three-atom ring was ca. 160 kJ/mol higher than the energy of its initial point.

It should be explicitly indicated that this value represented the difference in the electronic energies rather than the difference in Gibbs free energies, and therefore it could serve only as a rough estimate of the principal possibility of the Baeyer–Villiger pathway for this process. The value of 160 kJ/mol was quite a high barrier, which was significantly higher than the typical values of Baeyer–Villiger reaction barriers reported previously (e.g., 84 kJ/mol for furfural oxidation by H_2_O_2_ of sulfated zirconia [71]). The reaction rate, k, decreased exponentially with the increase in the barrier height, $\Delta G^{‡}$, according to the Eyring equation, Equation (2) [72].
(2)$$k=\frac{\kappa k_{B}T}{h}e^{\frac{-\Delta G^{‡}}{RT}}$$
where $\Delta G^{‡}$ is the Gibbs activation energy, κ is the transition coefficient and $k_{B}$ and h are the Bolzmann and Plank constants, respectively. Assuming κ=1, and considering the studied elementary step to be a first-order process, we can estimate the half-life value for the 160 kJ/mol barrier and 293 K as ca. 120 mln. years. On the contrary, considering 24 h as the maximal practically acceptable half-life, one can estimate the corresponding $\Delta G^{‡}$ as no more than ca. 100 kJ/mol.

In order to elucidate if there could be other reaction paths with lower energy barriers, we applied a search of the transition state between the two considered geometries via a Climbing Image Nudged Elastic Band (NEB-CI) procedure (see the Experimental part for details). It worth noting that, contrary to the RSS procedure, NEB-CI did not rely on any assumptions regarding the reaction path. Indeed, the path with a barrier of ca. 50 kJ/mol was found (the geometry of the transition state estimation is presented in Figure 17). This pathway implies breaking the C–C bond within the –(O–)(HOO)C–CH(OH)– fragment followed by breaking the O–O bond in the bound peroxide fragment and finally resulted in the formation of the ester and the expulsion of the OH^−^ anion.

The barrier of 50 kJ/mol corresponded to the half-life of the reaction step ca. 100 µs; this small value contradicted with the much slower experimentally observed process. The difference can either be explained by (i) a too rough estimation of the value, or (ii) the low content of deprotonated peroxide and activated benzoin in the system. The latter reason was consistent with the needs of H_2_O_2_ deprotonation as well as the decoordination of acetonitrile from the Cu^II^ ion prior to the reaction, which was modeled.

A similar calculation was performed for the modeling of the process of the bromo-substituted benzoin oxidation. Such a calculation yielded the same (within error) barrier height as for the unsubstituted benzoin. This result was in line with the experimental data (vide supra), which exhibited a difference in the reaction rates of less than 2 times (such a difference in the reaction rates corresponded to a ca. 2 kJ/mol difference in the activation barrier heights, which was too low value for reliable determination within the considered calculation setup).

## 3. Experimental and Computational Details

### 3.1. Materials and Methods

Commercially available reagents and solvents were supplied by UkrOrgSintez (Ukraine) and used without further purification. Metallic copper (99.9%, Khimlaborreaktiv, Ukraine) was used without further purification. Benzoin (98%, UkrOrgSintez) was purified by recrystallization from ethanol before use. Substituted benzoins were synthesized by the standard method from the corresponding aldehydes; NaCN was used as the condensation catalyst in the case of *p*-bromobenzaldehyde and 2-naphthaldehyde, while KCN was used in the case of *p*-methoxybenzaldehyde [73].

C,H,N analysis was performed using a Carlo Erba 1106 instrument. ^1^H NMR spectra were measured using a Bruker Advance 300 spectrometer. Powder X-ray diffraction measurements were performed using a Bruker D8 Advance diffractometer with CuKα radiation (λ = 1.54056 Å). TEM measurements were performed using a PEM-125K instrument (SELMI, Ukraine) operating at a 100 kV acceleration voltage. Samples were suspended in methanol upon ultrasonic irradiation for 1 min, and a drop of the suspension was applied onto a Cu grid (300 mesh) covered by a film of amorphous carbon immediately after the end of the ultrasonic treatment. N_2_ sorption experiments were carried out using a Sorptomatic-1990 instrument. Prior to the measurements, the sample was desolvated in a 10^−2^ Torr vacuum at 150 °C. Electronic adsorption spectra were measured using a Carl Zeiss Specord 210 spectrophotometer.

### 3.2. Synthesis of HKUST-1

Nanostructured HKUST-1 was prepared by the electrochemical dissolving of a Cu anode in a solution of H_3_btc, as described in the Supporting Materials, following a modification of the previously reported method [54].

### 3.3. Oxidation of Benzoin and Substituted Benzoins

In order to carry out the experiments in pseudo-zero order conditions of hydrogen peroxide, the reaction was performed in large excess of the latter. Thus, the influence of the H_2_O_2_ concentration change (consumption of H_2_O_2_) on the reaction rate could be neglected.

The typical procedure was as follows: 1.82 mmol of benzoin was dissolved in 36 mL of acetonitrile (c = 0.05 M) and 5 mole % of the catalyst was added (counting per 1 Cu^2+^ ion). Then, a 100-fold excess of H_2_O_2_ was added (as a 35% water solution). The reaction was carried out under intense stirring for 3 h at 20 °C, and probes were taken at certain time periods (for probe taking, the stirring was not stopped, and the effective concentration of the catalyst did not change). For the isolation of the products from the probes, the reaction was stopped by the addition of an excess of a saturated solution of Na_2_SO_3_ for the decomposition of H_2_O_2_. Then, the acetonitrile was evaporated, and the remaining water solution was extracted 3 times with ethylacetate. The extract was dried over Na_2_SO_4_ and evaporated. The remaining mixture was analyzed by NMR (the ratio of the starting compound and the product was determined by the ratio of signal intensities (Appendix A); in some cases, 1,3,5-trimethoxybenzene was added for calibration).

In the case of 1,2-bis(2-naphtyl)-2-hydroxyethanone, a mixture of acetonitrile and DMF (5:1 by volume) was used as solvent because of the low solubility of this compound in MeCN.

For the determination of the influence of the initial concentrations of the reagents (benzoin, H_2_O_2_, catalyst) on the process, the concentration of one of the reagents was changed, as described in the text of the paper.

In all the experiments, new samples of HKUST-1 were used. It was found that the repeated use of the catalyst led to a poor reproducibility of the initial reaction rates; in all cases, the reaction rates in the presence of the recycled catalyst were lower compared to the values achieved in the presence of new samples of the catalyst. Such a slowing down of the reaction could be explained by the occlusion of the pores and blocking of the reaction sites by the reagents and products.

### 3.4. Studies of the Adsorption of Reagents on HKUST-1

Experiments on the sorption isotherm and sorption kinetic measurements were carried out at 20 °C as previously described [14]. Concentrations of the compounds in solutions were measured by UV spectroscopy. The mixture of acetonitrile and water (corresponding to water content, added in 35% water solution of H_2_O_2_) was used as solvent instead of the mixture of acetonitrile, water and H_2_O_2_, because the spectra in presence of H_2_O_2_ had very low quality.

### 3.5. DFT Calculations

All DFT calculations were performed via the ORCA 5.0.3 software (Frank Neese et al., Max-Planck-Institut für Kohlenforschung, Mülheim an der Ruhr, Germany) [74,75,76], employing the def2-SVP basis set [77,78] (together with the def2/J auxiliary basis [79] for speeding up the calculations by the resolution of identity approach) and the PBE exchange-correlation functional [80]. D3 correction for the van der Waals forces [81], together with the Becke–Johnson damping function [82], were applied. The effects of the medium (acetonitrile) were taken into account via the CPCM implicit solvation scheme [83]. Paramagnetic species were modeled via spin-unrestricted DFT.

In order to avoid modeling with periodic boundary conditions and for reducing the required computational effort, a Cu_2_(CH_3_CO_2_)_4_ species (in the triplet state) was taken as a model for an active site in the HKUST-1 coordination polymer.

Initial geometries of the species of interest, unless derived from the previous calculations, were built using the Avogadro 1.1.1 software (Marcus D. Hanwell et al., USA) [84] and optimized with UFF forcefield [85], as implemented in OpenBabel library [86], employed by the Avogadro editor.

In order to follow a hypothesized reaction coordinate, the relaxed surface scan (RSS) procedure, as implemented in ORCA, was applied. The scan was started from the equilibrium geometry. The degree of freedom (within this work, the distance between a pair of atoms) of interest was changed stepwise (with a 0.1 Å step). At each step, a constrained geometry optimization was performed, fixing the value of the scanned distance and allowing all other coordinates to relax toward the minimal energy. After the final step, the geometry was additionally optimized without any constraints to gain the equilibrium geometry of the further intermediate.

In order to estimate the structure and the energy of the transition state, the climbing image nudged elastic band (NEB-CI) procedure with 12 intermediate geometries (“images”) was applied [87,88,89]. Within this procedure, the minimal energy path between two given equilibrium geometries was searched, and one of the images of the path with the highest energy found after a preliminary optimization was pushed toward the maximum energy along the path (the climbing image, representing the estimation of the transition state).

The atomic charges were derived from the electronic densities calculated by DFT via the MultiWFN software (Tian Lu, Beijing Kein Research Center for Natural Sciences, Beijing, China) [90] according to the following schemes: atomic dipole moment corrected Hirshfeld population (ADCH) [91], CHELPG [92], (original) Hirshfeld population [93], Becke atomic charge with atomic dipole moment correction [91,94] and Mulliken population [95].

## 4. Conclusions

It was found that the dependency of the average initial rate of benzoin oxidation by H_2_O_2_ in the presence of HKUST-1 on the benzoin concentration passed through a maximum. Such a complex dependency can be explained by the balance between a more efficient catalytic oxidation and a more efficient blocking of the catalytically active cites (which prevented access by H_2_O_2_) caused by benzoin adsorption. Besides this finding, an increase in the efficient concentration of HKUST-1 to a certain value also led to a decrease in the average initial rate of benzoin oxidation, which can also be explained by the adsorption of benzoin. Such adsorption led to a decrease in its concentration in the solution as well as to blocking of the active cites of the HKUST-1. The significant adsorption of benzoin on the HKUST-1 was confirmed experimentally.

It was found that average initial rate of oxidation of a series of substituted benzoins by H_2_O_2_ depended on the electronic structure of the molecule and did not correlate with their size. This observation provided evidence for the accessibility of the active sites of the catalyst. The dependency of the oxidation rate on the electronic structure of the reagents could be an argument for Baeyer–Villiger mechanism of oxidation in contrast to a radical-based one.

The possible mechanism of oxidation, involving an attack of the HOO^−^ ion on the C atom of activate carbonyl group, was confirmed by DFT calculations. The reaction path with a barrier of ca. 50 kJ/mol was found, which implied the breaking of the C–C bond within the –(O–)(HOO)C–CH(OH)– fragment followed by the breaking of the O–O bond in the bound peroxide fragment and finally resulted in the formation of the ester and the expulsion of the OH^−^ anion.

In our opinion, the results of this study can be useful both for a deeper understanding of the specific features of the oxidation of organic compounds catalyzed by porous systems (especially the interplay between oxidation and adsorption) as well as a deeper understanding of the mechanisms of the oxidation of carbonyl compounds by H_2_O_2_.

## Figures and Tables

**Figure 1 molecules-28-00747-f001:**
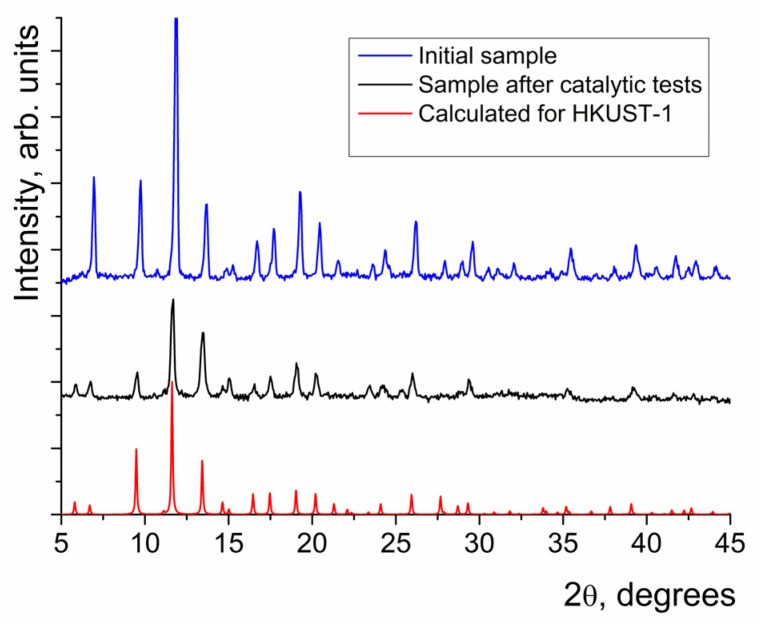
The experimental XRD patterns obtained for HKUST-1 after synthesis and after catalytic tests and a theoretically calculated one for HKUST-1.

**Figure 2 molecules-28-00747-f002:**
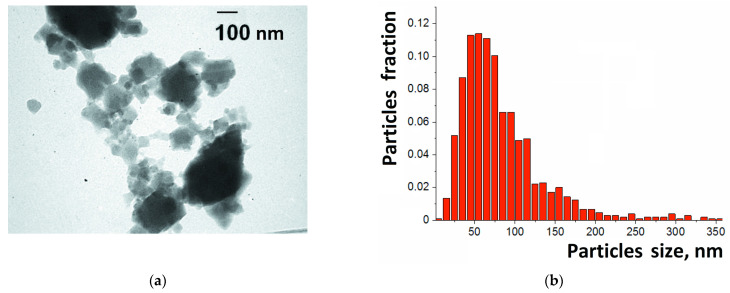
TEM image (**a**) and particle size distribution (**b**) for HKUST-1 used in this study.

**Figure 3 molecules-28-00747-f003:**
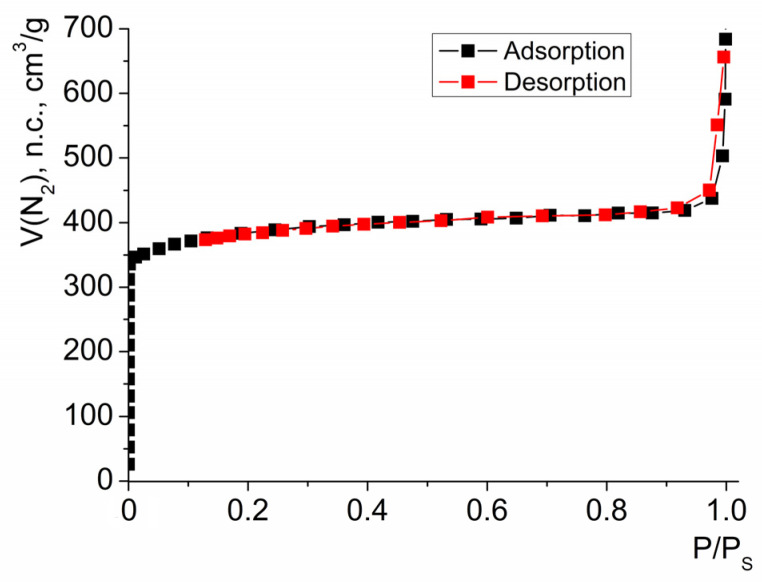
An isotherm of nitrogen sorption by HKUST-1 used in this study.

**Figure 4 molecules-28-00747-f004:**
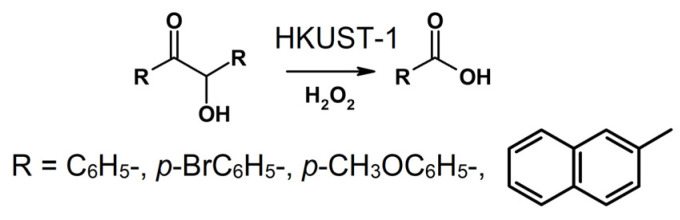
Catalytic oxidation of benzoin and substituted benzoins with excess of H_2_O_2_ in the presence of HKUST-1.

**Figure 5 molecules-28-00747-f005:**
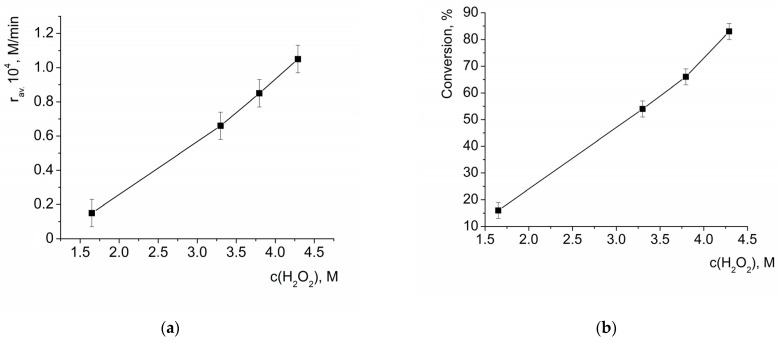
Dependency of the average initial rate of benzoin oxidation by H_2_O_2_ in the presence of HKUST-1 (**a**) and conversion of benzoin after 3 h of reaction (**b**) on the initial concentration of H_2_O_2_. c(benzoin) = 0.033 M, c_eff_.(HKUST-1) = 0.001 M.

**Figure 6 molecules-28-00747-f006:**
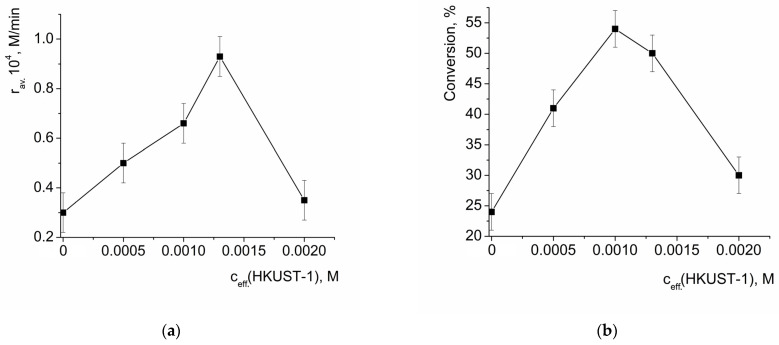
Dependency of the average initial reaction rate of benzoin oxidation by H_2_O_2_ (**a**) and benzoin conversion after 3 h of reaction (**b**) on the initial effective concentration of HKUST-1. c(benzoin) = 0.033 M, c(H_2_O_2_) = 3.3 M.

**Figure 7 molecules-28-00747-f007:**
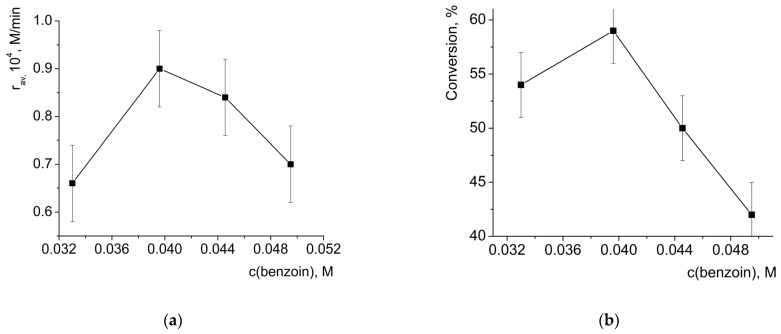
Dependency of the average initial reaction rate of benzoin oxidation by H_2_O_2_ (**a**) and benzoin conversion after 3 h of reaction (**b**) on the initial concentration of benzoin. c_eff._(HKUST-1) = 0.001 M, c(H_2_O_2_) = 3.3 M.

**Figure 8 molecules-28-00747-f008:**
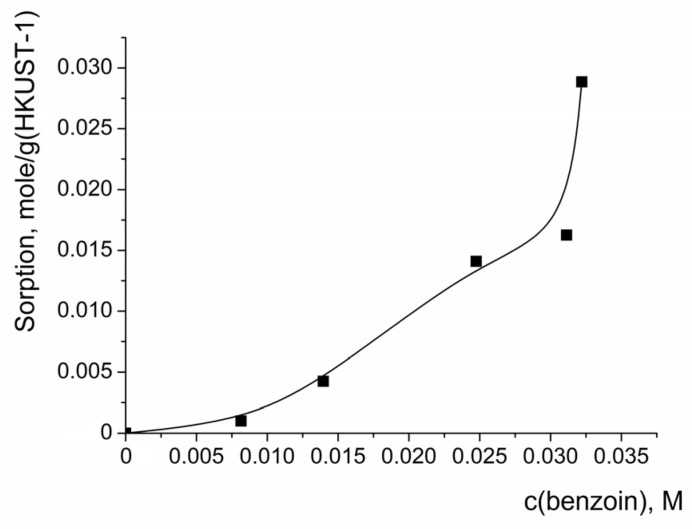
Isotherm of benzoin adsorption on HKUST-1 from solution in a mixture of acetonitrile and water (corresponding to solvent composition in the reaction mixture in the oxidation experiments, where water was added as a component of a 35% water solution of H_2_O_2_). Solid line in the graph is to guide the eye. C(benzoin) is equilibrium concentration.

**Figure 9 molecules-28-00747-f009:**
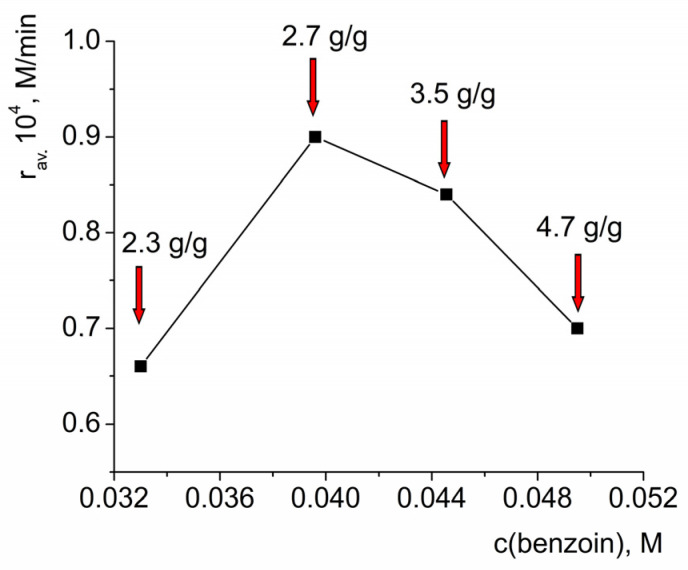
Estimation of the quantity of adsorbed benzoin per 1 g of the catalyst at initial concentration of benzoin vs. average initial oxidation rate. c_eff._(HKUST-1) = 0.001 M, c(H_2_O_2_) = 3.3 M. Solid line in the graph is to guide the eye.

**Figure 10 molecules-28-00747-f010:**
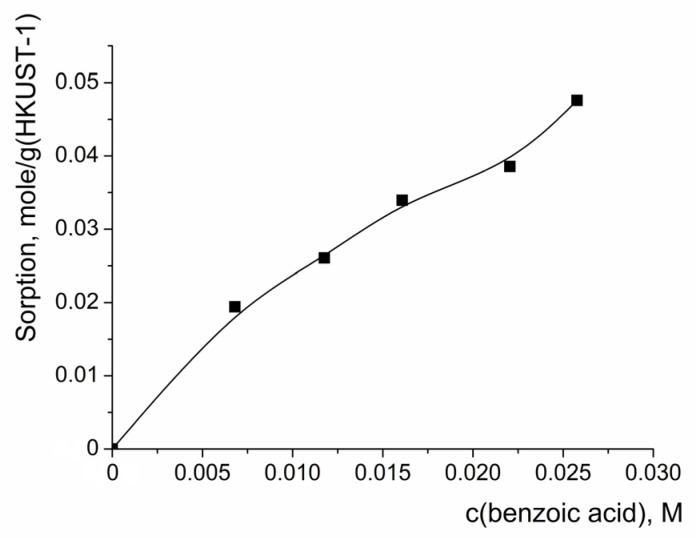
Isotherm of benzoic acid adsorption on HKUST-1 sample. Solid line in the graph is to guide the eye. C(benzoic acid) is equilibrium concentration.

**Figure 11 molecules-28-00747-f011:**
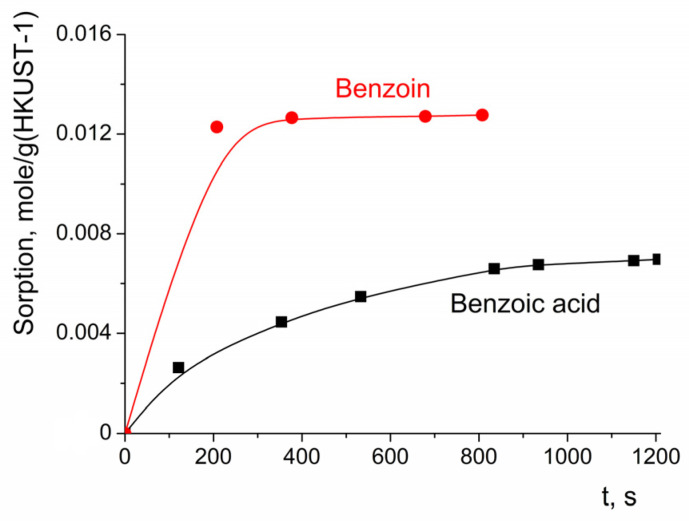
Dependency of benzoin and benzoic acid sorption on HKUST-1 on time. c_0_(benzoin) = 0.0165 M; c_0_(benzoic acid) = 0.0165 M.

**Figure 12 molecules-28-00747-f012:**
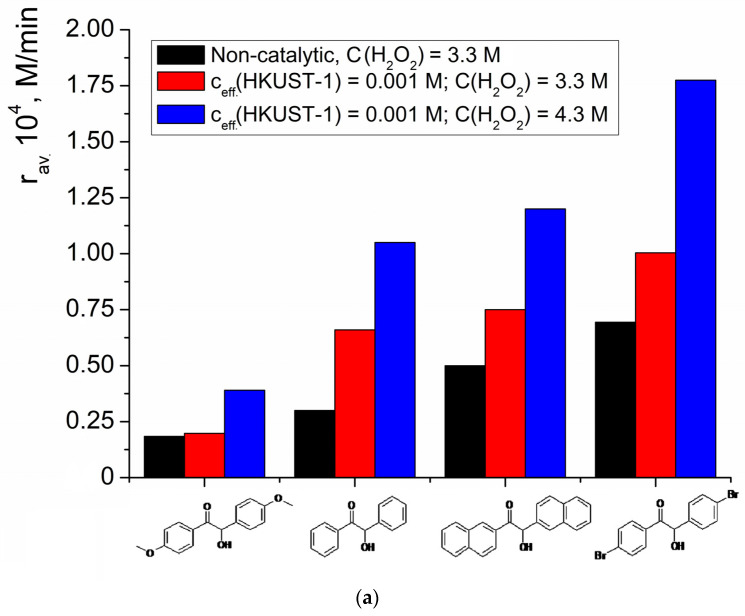
Comparison of the average initial rate (**a**) and the conversions after 3 h of reaction (**b**) in the reactions of benzoin and substituted benzoin oxidation by hydrogen peroxide without catalyst and in the presence of HKUST-1 (sum of catalytic and non-catalytic reactions). c(benzoin) = 0.033 M, c_eff_.(HKUST-1) and c(H_2_O_2_) are shown in the graph.

**Figure 13 molecules-28-00747-f013:**
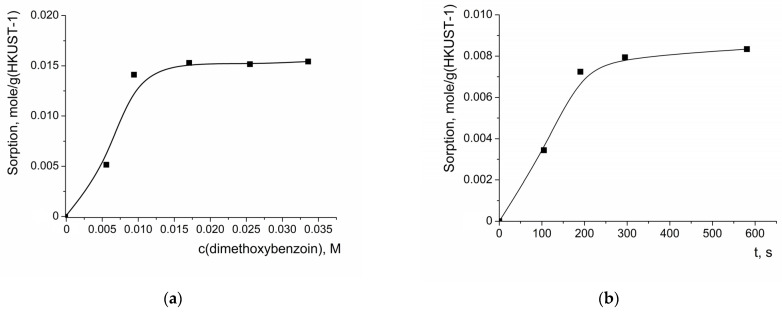
Isotherm of 4,4-dimethoxybenzoin adsorption on HKUST-1 (**a**) and kinetic curve of this compound adsorption (c_0_ = 0.0165 M) (**b**).

**Figure 14 molecules-28-00747-f014:**
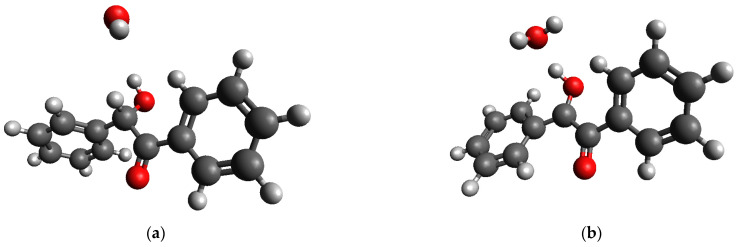
(**a**) Initial and (**b**) equilibrium (optimized by DFT) geometries of {benzoin+ •OH} species.

**Figure 15 molecules-28-00747-f015:**
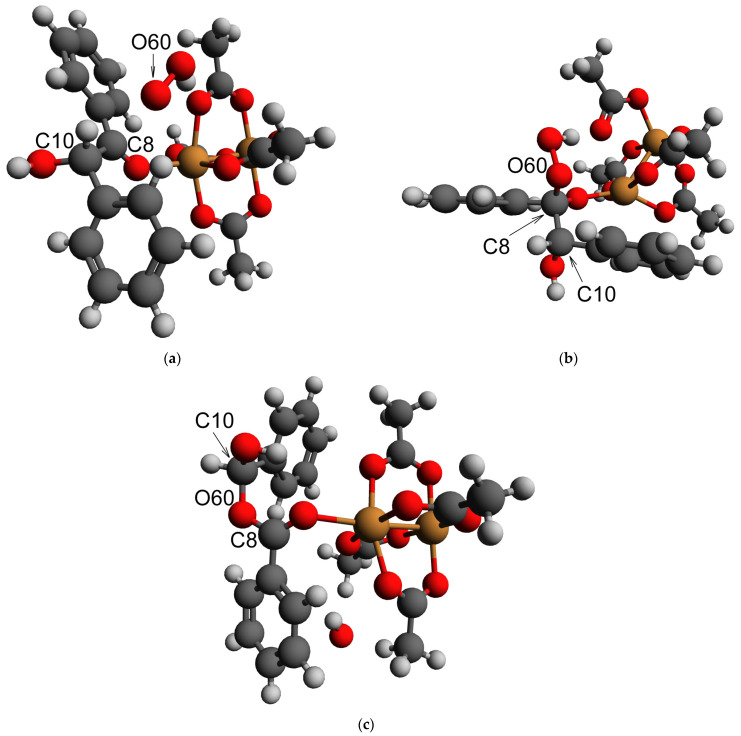
Equilibrium geometries (obtained by DFT) of the intermediates of the supposed Baeyer–Villiger oxidation: (**a**) before HOO^−^ attack; (**b**) after HOO^−^ attack nudged via RSS procedure; (**c**) after the rearrangement nudged via RSS procedure. Orange: Cu, red: O; dark grey: C; light grey: H.

**Figure 16 molecules-28-00747-f016:**
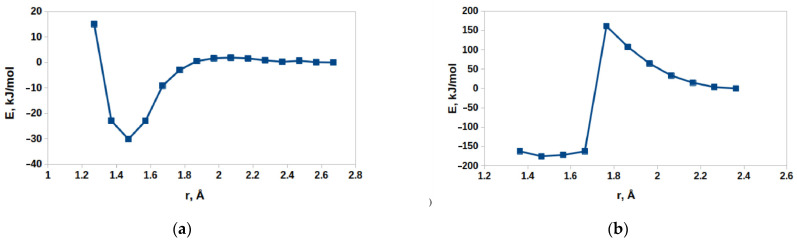
RSS energy profiles of (**a**) the HOO^−^ attack and (**b**) the rearrangement. Each point depicts the energy of the species yielded by the constrained optimization with the corresponding constraint value imposed to the scanned distance. The connecting lines were added solely for guiding the eye.

**Figure 17 molecules-28-00747-f017:**
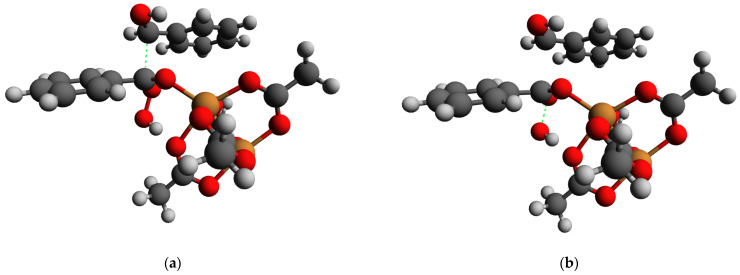
(**a**) The geometry of the transition state of the Baeyer–Villiger reaction pathway of the benzoin oxidation estimated by the NEB climbing image; (**b**) the image representing the result of the O–O bond cleavage. Orange: Cu, red: O; dark grey: C; light grey: H. Green dotted lines indicate positions of broken bonds.

## Data Availability

The data presented in this study are available on request from the corresponding author.

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
