# Peer review of "Catalytic Oxidation of Benzoins by Hydrogen Peroxide on Nanosized HKUST-1: Influence of Substituents on the Reaction Rates and DFT Modeling of the Reaction Path"

_molecules, 2023, doi:10.3390/molecules28020747_

Round 1
Reviewer 1 Report
- To ensure the heterogeneous nature of the catalysis, please show the results of the hot filtration test, and do quantitative analysis (not only qualitative) of the Cu2+ present in the solution after filtering the solid.
- Line 190 “ dont close
- Line 203. The , after solution should be a .
- Line 204 the can after decomposition should be eliminated
- Line 289 the first the should be to
- The authors should add exepriments regarding the reusability of the catalyst.
- Have the authors tried another type of peroxides or another type of oxidants for the reaction? The use of H2O2 3.3 or 4.3 M seems not very friendly for real applications.
- In the experimental section the authors say: In the case of 1,2-bis(2-naphtyl)-2-hydroxyethanone the mixture of acetonitrile and DMF (5:1 by volume) was used as solvent because of low solubility of this compound in MeCN. Have the authors tried the stability of the HKUST-1 catalyst in this mixture acetonitrile and DMF (5:1 by volume) as solvent? Plesae, provide experimental evidences.
Author Response
Reviewer 1
We thank the Reviewer for his/her valuable comments and suggestions.
COMMENT
- To ensure the heterogeneous nature of the catalysis, please show the results of the hot filtration test, and do quantitative analysis (not only qualitative) of the Cu2+ present in the solution after filtering the solid.
REPLY
We estimated concentration of Cu2+ ions in the selected reaction mixtures after catalyst filtration tests by atomic adsorption spectroscopy (actually, it was not "hot filtration", because the catalytic experiments were carried out at 20 ºC). These mixtures included those with the lowest effective concentration of HKUST-1 (i.e. 5·10-4 M) and the highest effective concentration of HKUST-1 (i.e. 2·10-3 M). In all cases concentration of Cu2+ ions was less that 1·10-7 M. The comment was added to the text of the manuscript.
COMMENT
- Line 190 “ dont close
REPLY
This was corrected
COMMENT
- Line 203. The , after solution should be a .
REPLY
This was corrected.
COMMENT
- Line 204 the can after decomposition should be eliminated
REPLY
This was corrected.
COMMENT
- Line 289 the first the should be to
REPLY
This was corrected, more significant changes were made in the sentence.
COMMENT
- The authors should add exepriments regarding the reusability of the catalyst.
REPLY
We repeated several experiments on the reusability of the catalyst and in all cases the results (reaction rates and conversion values of the starting compounds) were poorly reproducible. In all cases, when "used" catalysts were taken, the initial reaction rates were lower compared to the experiments with "fresh" catalysts. We explain such poor reproducibility by occlusion of pores and blocking of the active sites with the reagents and products. The comment was added to the text (in Experimental section).
This finding does not change the conclusions of the study, which concern analysis of the factors, that control the reaction rate in the process of catalytic oxidation of benzoins.
COMMENT
- Have the authors tried another type of peroxides or another type of oxidants for the reaction? The use of H2O2 3.3 or 4.3 M seems not very friendly for real applications.
REPLY
In this study we did not use another oxidants. We have experience of organic compounds (alcohols) oxidation with tert-butylperoxide or m-chloroperoxobenzoic acid, and we are preparing papers on these results now. In our experience, even concentrated H2O2 (such as 3 – 5 M) is much more friendly and more ecologically safe compared to other organic and inorganic peroxides or such oxidants as KMnO4, K2Cr2O7, OsO4, because hydrogen peroxide produces only water as the product, and the excess of H2O2 can be easily decomposed (again, to water and oxygen). No other oxidant is so ecologically friendly (maybe except O2, but use of O2 often requires more complex and sometimes more toxic catalysts).
Work with 3–5 M solutions of H2O2 in solvents, which can't form organic peroxides, is safe. For example, 35 % solution of H2O2 (ca. 12 M) is common commercial reagent for treating water in swimming pools for people, it can be easily transported in plastic vessels without special precautions and handled even for non-professional household use in small swimming pools.
COMMENT
- In the experimental section the authors say: In the case of 1,2-bis(2-naphtyl)-2-hydroxyethanone the mixture of acetonitrile and DMF (5:1 by volume) was used as solvent because of low solubility of this compound in MeCN. Have the authors tried the stability of the HKUST-1 catalyst in this mixture acetonitrile and DMF (5:1 by volume) as solvent? Plesae, provide experimental evidences.
REPLY
The catalyst is stable in the mixture of DMF and acetonitrile. In solutions, obtained after catalyst filtration tests for these reaction mixtures, concentration of Cu2+ was less than 10-7 M. Also please note, that many methods for synthesis of HKUST-1 involve DMF as solvent, and DMF is also used for "washing out" of HKUST-1 samples for their post-synthetic purification from the residues of benzenetricarboxylic acid.
Reviewer 2 Report
In this work, authors performed a combined experimental and computational study on the oxidation of benzoin by H2O2. They calibrate the influence of oxidant H2O2 and supposed catalyst HKUST-1. While they found reaction rate is first order on H2O2, but the role of HKUST-1 is elusive. DFT calculations provide some insights however not closely connected with experimental observation.
1. I guess HKUST-1 is catalyst in this reaction. If this holds true, there are not clear evidence to support this role. Moreover, authors claimed that benzoin adsorption on HKUST-1 wil poison reaction. how a catalyst to work if reactant did not bind on its surface?
2. Why need so much excess H2O2?
Author Response
Reviewer 2
We thank the Reviewer for his/her valuable comments and suggestions.
COMMENT
In this work, authors performed a combined experimental and computational study on the oxidation of benzoin by H2O2. They calibrate the influence of oxidant H2O2 and supposed catalyst HKUST-1. While they found reaction rate is first order on H2O2, but the role of HKUST-1 is elusive. DFT calculations provide some insights however not closely connected with experimental observation.
- I guess HKUST-1 is catalyst in this reaction. If this holds true, there are not clear evidence to support this role. Moreover, authors claimed that benzoin adsorption on HKUST-1 wil poison reaction. how a catalyst to work if reactant did not bind on its surface?
REPLY
The role of HKUST-1 as the catalyst was confirmed by comparison of the reaction rates at presence of different loadings of HKUST-1 and without HKUST-1, as well as by filtration tests. DFT calculations are also consistent with the role of HKUST-1 as the catalyst.
The reactant efficiently bound on the surface of the catalyst - high sorption ability of HKUST-1 towards benzoin was the evidence for binding of the latter. We explain slowing down of the reaction at high concentrations of the reagent by the consequence of its binding (slow desorption and restricted access of H2O2, as well as slow molecules exchange in close proximity to the active sites). It is not poisoning in common meaning.
Regarding Referee's comment that DFT calculations were not closely connected with experimental observations, we carried out DFT calculations in order to confirm that Bayer-Villiger mechanism could be realistic in this case. Indeed, the realistic reaction path with the barrier of ca. 50 kJ/mol was found. In the revised version we extended this section in order to make it more clear. We also performed similar calculation for Br-substituted benzoin and got very similar results; in our opinion, this new calculation strengthens the reliability of our conclusion.
COMMENT
- Why need so much excess H2O2?
REPLY
We took large excess of H2O2 for two reasons: in order to be able to neglect change of the concentration of H2O2 in course of the reaction, and to force formation of one reaction product - benzoic acid (not a mixture with benzil) - for simplification of kinetic studies.
Reviewer 3 Report
Darya V. Yurchenko et al. reported the oxidation of a series of benzoins R-C(=O)-CH(OH)-R, where R = phenyl, 4-methoxyphenyl, 4-bromophenyl and 2-naphthyl, by hydrogen peroxide at presence of nanostructured HKUST-1 (in suspension in acetonitrile/water mixture) was studied. The initial average reaction rates were experimentally determined at different concentrations of benzoin, H2O2 and effective concentration of HKUST-1. The isotherms of benzoin, dimethoxybenzoin and benzoic acid sorption on HKUST-1, as well as sorption kinetic curves, were measured. The electronic effect of the substituent in benzoin had significant influence on the reaction rate, while no relation between the size of the substrate molecule and the rate of its oxidation was found. DFT modeling the reaction mechanism was also carried out.
Recommendations: In this manuscript, many grammatical and scientific mistakes are observed. So, I recommend this manuscript for publishing in the Journal of molecules with following suggestions:
1. Author must maintain line spacing of 1.5 throughout the manuscript.
2. The whole manuscript is written in Palatino Linotype writing style but the size of the writing is not same throughout the manuscript, abstract and keywords have size 9, while remaining article is written with the size 10. So please maintain the same size of writing throughout the manuscript.
3. In abstract portion, the sentence “DFT modeling the reaction mechanism was carried out” is not looking meaningful. So please replace it with a suitable sentence.
4. It is observed that, some sentences in the manuscript are too much long. Long and convoluted sentences are harder to understand and affect the readability and comprehension. Use moderate size sentences uniformly throughout the manuscript.
5. In the first sentence of introduction “In the last decades a lot of attention has been paid to heterogeneous catalyst of organic compounds oxidation reactions” the words “oxidation reactions” are not at right place. So reset this sentence again in a proper way.
6. In introduction, before reference number 13 and 14 the word “cites” is used. Replace it with the word “sites” because it’s a spelling mistake.
7. Hyphen must be used while writing the words nanoparticles and nanostructures such as nano-particles, nano-structures.
8. In the second paragraph of introduction the word “capacity of use” is utilized. Please replace it with the correct one “capacity to use”.
9. Introduction part is written in present tense while at some places past tense is utilized. So, maintain the same grammatical style throughout the introduction.
10. In the last sentence of introduction use the word “modeling of reaction” instead of “modeling reaction”.
11. In the description of figure 2b “sizes of particles were measured” is written, kindly replace it with the suitable one “size of particles was measured”.
12. Italicize the units and the words like “via, etc ,i.e, vs.” throughout the manuscript.
13. Write “in figure, in graph” instead of “on figure, on graph” while mentioning the figures and graphs throughout the manuscript.
14. In the paragraph below the figure 7 the words “negligible and small” used at the same time. Use any one of them or replace it with “negligibly small”.
15. In figure 8 c(benzoin) is written while in the description capital C is used before bezoin i.e., C(benzoin). Adjust them in similar manner at both places.
16. Below the equation 1 word “the” is written two times accordingly. Use it only one time.
17. In the sentence below the figure 13 in second paragraph “seem to be not the” is used. Write it in a proper way i.e., “seem not to be”.
18. In figure 15 and 17 the atom numbers are not clear. So, add these figures again in the manuscript with easily visible numbering.
19. After the reference 64 in the description of figure 16a the abbreviation of Relaxed Surface Scan is not enclosed in the bracket properly i.e., “(RSS”. Mention it correctly as (RSS).
20. In results and discussion Relaxed Surface Scan (RSS), Climbing Image Nudged Elastic Band (NEB-CI) are not italic while in DFT calculations “Relaxed Surface Scan, Climbing Image Nudged Elastic Band” are italic. Please maintain the same writing style throughout the manuscript.
21. In the DFT portion, kindly cite the following articles;
a. https://doi.org/10.1016/j.molstruc.2020.127803.
b. https://doi.org/10.1016/j.molstruc.2019.127633.
c. https://doi.org/10.1016/j.molstruc.2019.127438
22. In the first paragraph of conclusion the sentence “Besides this finding, increase of the efficient concentration of HKUST-1 to certain value also led to decrease of the average rate of benzoin oxidation, which also can be explained by adsorption of benzoin and associated decrease of its concentration in solution along with blocking of the active cites of HKUST-1.” Is not proper, write “which can also be” instead of “which also can be” and also replace the word associated with another suitable word.
23. The heading “Conclusions” must be replaced with “Conclusion”.
Author Response
Reviewer 3
We thank the Reviewer for his/her valuable comments and suggestions.
COMMENT
- Author must maintain line spacing of 1.5 throughout the manuscript.
REPLY
We used template, provided by the Editorial office, and performed formatting using the Style option, embedded in the template. Before submission of the revised version, we checked the manuscript again.
COMMENT
- The whole manuscript is written in Palatino Linotype writing style but the size of the writing is not same throughout the manuscript, abstract and keywords have size 9, while remaining article is written with the size 10. So please maintain the same size of writing throughout the manuscript.
REPLY
We used template, provided by the Editorial office, and performed formatting using the Style option, embedded in the template. These font sizes were provided by the template. Before submission of the revised version, we checked the manuscript again.
COMMENT
- In abstract portion, the sentence “DFT modeling the reaction mechanism was carried out” is not looking meaningful. So please replace it with a suitable sentence.
REPLY
We agree with the Referee and this phrase was changed. In the revised version we wrote in abstract section, what exactly was done by DFT modeling.
COMMENT
- It is observed that, some sentences in the manuscript are too much long. Long and convoluted sentences are harder to understand and affect the readability and comprehension. Use moderate size sentences uniformly throughout the manuscript.
REPLY
We revised the whole text of the manuscript and changed some sentences, especially those which were too long.
COMMENT
- In the first sentence of introduction “In the last decades a lot of attention has been paid to heterogeneous catalyst of organic compounds oxidation reactions” the words “oxidation reactions” are not at right place. So reset this sentence again in a proper way.
REPLY
This sentence was changed.
COMMENT
- In introduction, before reference number 13 and 14 the word “cites” is used. Replace it with the word “sites” because it’s a spelling mistake.
REPLY
It was our mistake, we thank the Referee. It was corrected.
COMMENT
- Hyphen must be used while writing the words nanoparticles and nanostructures such as nano-particles, nano-structures.
REPLY
We know that there are two styles of writing the words with "nano" prefix, i.e., for example, nanoparticles and nano-particles. We prefer to use the first style, because our analysis of the literature showed that it is more widespread.
COMMENT
- In the second paragraph of introduction the word “capacity of use” is utilized. Please replace it with the correct one “capacity to use”.
REPLY
This was corrected.
COMMENT
- Introduction part is written in present tense while at some places past tense is utilized. So, maintain the same grammatical style throughout the introduction.
REPLY
We revised the introduction. Actually, information in the first three paragraphs is mainly presented in the present tense, because it is common knowledge. In the next paragraphs we present information, which was found in certain studies or present information about our decisions (the aim of the study, selection of objects, etc.), and this information is presented in the past. We corrected some places to make these paragraphs more uniform from the viewpoint of grammatical style.
COMMENT
- In the last sentence of introduction use the word “modeling of reaction” instead of “modeling reaction”.
REPLY
This was corrected.
COMMENT
- In the description of figure 2b “sizes of particles were measured” is written, kindly replace it with the suitable one “size of particles was measured”.
REPLY
This was corrected.
COMMENT
- Italicize the units and the words like “via, etc ,i.e, vs.” throughout the manuscript.
REPLY
This was corrected.
COMMENT
- Write “in figure, in graph” instead of “on figure, on graph” while mentioning the figures and graphs throughout the manuscript.
REPLY
This was corrected.
COMMENT
- In the paragraph below the figure 7 the words “negligible and small” used at the same time. Use any one of them or replace it with “negligibly small”.
REPLY
This was corrected.
COMMENT
- In figure 8 c(benzoin) is written while in the description capital C is used before bezoin i.e., C(benzoin). Adjust them in similar manner at both places.
REPLY
This was corrected.
COMMENT
- Below the equation 1 word “the” is written two times accordingly. Use it only one time.
REPLY
This was corrected.
COMMENT
- In the sentence below the figure 13 in second paragraph “seem to be not the” is used. Write it in a proper way i.e., “seem not to be”.
REPLY
This was corrected.
COMMENT
- In figure 15 and 17 the atom numbers are not clear. So, add these figures again in the manuscript with easily visible numbering.
REPLY
We changed these figures and left atoms labels only for those atoms, which are mentioned in the text.
COMMENT
- After the reference 64 in the description of figure 16a the abbreviation of Relaxed Surface Scan is not enclosed in the bracket properly i.e., “(RSS”. Mention it correctly as (RSS).
REPLY
In this place the whole comment "(RSS, see Experimental part for the details of the approach)" was included in the brackets. We prefer to leave it in one pair of brackets, instead of writing like "(RSS), (see Experimental part for the details of the approach)".
COMMENT
- In results and discussion Relaxed Surface Scan (RSS), Climbing Image Nudged Elastic Band (NEB-CI) are not italic while in DFT calculations “Relaxed Surface Scan, Climbing Image Nudged Elastic Band” are italic. Please maintain the same writing style throughout the manuscript.
REPLY
This was corrected.
COMMENT
- In the DFT portion, kindly cite the following articles; a. https://doi.org/10.1016/j.molstruc.2020.127803. b. https://doi.org/10.1016/j.molstruc.2019.127633. c. https://doi.org/10.1016/j.molstruc.2019.127438
REPLY
We added these references (Refs. 66-68 in the revised version)
COMMENT
- In the first paragraph of conclusion the sentence “Besides this finding, increase of the efficient concentration of HKUST-1 to certain value also led to decrease of the average rate of benzoin oxidation, which also can be explained by adsorption of benzoin and associated decrease of its concentration in solution along with blocking of the active cites of HKUST-1.” Is not proper, write “which can also be” instead of “which also can be” and also replace the word associated with another suitable word.
REPLY
This was done. The sentence was split into two separate sentences.
COMMENT
- The heading “Conclusions” must be replaced with “Conclusion”.
REPLY
This was corrected.
Round 2
Reviewer 1 Report
No comments
Reviewer 3 Report
it is improved and can be consider it for publication